# Integrated Application of Rapeseed Cake and Green Manure Enhances Soil Nutrients and Microbial Communities in Tea Garden Soil

Haiping Fu [1,2,†], Huan Li [3,†], Peng Yin [2], Huiling Mei [4], Jianjie Li [4], Pinqian Zhou [1], Yuanjiang Wang [1], Qingping Ma [5], Anburaj Jeyaraj [5], Kuberan Thangaraj [5], Xuan Chen [5], Xinghui Li [5,*] and Guiyi Guo [2,*]

[1] Tea Research Institute, Hunan Academy of Agricultural Sciences, Changsha 410126, China; fuhaiping2010@126.com (H.F.); xniancier@163.com (P.Z.); chnwyj@163.com (Y.W.)
[2] Henan Key Laboratory of Tea Comprehensive Utilization in South Henan, Xinyang Agriculture and Forestry University, Xinyang 464000, China; 52greentea@sina.com
[3] Institute of Leisure Agriculture, Jiangsu Academy of Agricultural Sciences, Nanjing 210014, China; 20170032@jaas.ac.cn
[4] College of Resources and Environmental Sciences, Nanjing Agricultural University, Nanjing 210095, China; 2019203030@njau.edu.cn (H.M.); 2019203032@njau.edu.cn (J.L.)
[5] Tea Research Institute, Nanjing Agricultural University, Nanjing 210095, China; maqingpingtea@163.com (Q.M.); tku2010@gmail.com (A.J.); geneanbu1986@gmail.com (K.T.); chenxuan@njau.edu.cn (X.C.)
\* Correspondence: lxh@njau.edu.cn (X.L.); ggy6363@aliyun.com (G.G.); Tel.: +86-25-8439-6651 (X.L.); +86-376-668-7756 (G.G.)
† These authors have contributed equally to this study.

**Abstract:** (1) Aims: This study was aimed to investigate the effects of organic and inorganic fertilizer application on the soil nutrients and microbiota in tea garden soil. (2) Method: Illumina Hiseq sequencing technique was conducted to analyze the microbial diversity and density in different fertilizer-applied tea garden soil. (3) Results: The results showed that *Acidobacteria*, *Proteobacteria* and *Actinobacteria* were the predominant bacterial species observed in the tea garden soil. Besides, the relative abundance of *Basidiomycota*, *Ascomycota* and *Zygomycota* fungal species were higher in the tea garden soil. Correlation analysis revealed that *Acidibacter* and *Acidothermus* were significantly correlated with chemical properties (such as total organic carbon (TOC), total phosphorus (TP) and available phosphorus (AP) contents) of the tea garden soil. Furthermore, all these microbes were abundant in medium rapeseed cake (MRSC) + green manure (GM) treated tea garden soil. (4) Conclusion: Based on the obtained results, we conclude that the application of MRSC + GM could be a preferred fertilizer to increase the soil nutrients (TOC, TP and AP content) and microbial population in the tea garden soil.

**Keywords:** fertilizer; green manure; microbiota; rapeseed cake; soil nutrient



## 1. Introduction

Soil is a complicated ecosystem providing many environmental and agriculture services including food for almost every organism on earth [1]. The nitrogen fertilizer application causes complex effects on soil properties such as soil chemical, biological and physical conditions, and even on the degradation of the soil quality and reduction plant productivity [2]. Soil microorganisms play an important role in cycling nutrients and the energy flow [3], while bacteria and fungi have essential impacts on physical and chemical conditions of soil [4]. It is known that community structure, diversity and abundance of soil microorganisms are sensitive to the land usage and soil management [5]. Therefore, it is important to understand the dynamics of the soil microbial community.

Tea plants (*Camellia sinensis* (L.) O. Kuntze), a famous perennial evergreen woody economic crop in the world that grow well in acidic soils with pH range from four to six.

are widely distributed in tropical and subtropical places of China [6]. China is the largest tea producer in the world with $1.76 \times 10^6$ hectares of tea plant area and $1.94 \times 10^6$ ton of yield in 2013 [7]. However, soil deterioration has been observed in long-term monoculture of tea plants, which, in turn, has led to massive output reduction of tea [8]. Presently, a lot of nitrogen fertilizers which can reach 2000 kg N ha$^{-1}$ in some areas of China and Japan have been applied to stabilize the economic benefits of tea. Soil acidification is aggravated as a result of leaching of acidic ions [9]. Three elements have been considered to alter the soil quality of tea rhizosphere, including the changes in microbial community structure, soil physical and chemical properties [10]. Moreover, the deficiency of soil nutrient availability can seriously affect the physiological condition, vitality and community structure of rhizosphere microorganism in tea garden, which influence the yield and quality of tea.

In the present investigation, Illumina Hiseq sequencing approach was used to evaluate the abundance and the structure of microbiota in tea garden soil that was being exposed to different fertilizer applications over 3 years. Additionally, the soil nutrient parameters were also detected and analyzed to evaluate the effect of different fertilizer on soil properties and tea productivity and quality. Based on the results, a better fertilizer combination would be proposed for improving the ecological sustainability of tea orchard ecosystem.

## 2. Materials and Methods

### 2.1. Experiment Design and Fertilizer Treatment

The experiment was conducted in Gaoqiao of Hunan Province (113°19′ E, 28°29′ N, 68 MSL with an annual rain of 1380 mm) in 2015. *Camellia sinensis* cv. Fuding Dabai was planted in the experimental field. The field management, pest control and other management measures of each plot are the same to ensure the normal growth of tea trees. The soil samples used in this research are treated with the following treatments and the details, namely, zero fertilizer (CK), chemical fertilizer (CF, 450 kg/ha), rapeseed cake (RSC, 6750 kg/ha), RSC + CF (6750 + 180 kg/ha), low RSC + green manure (LRSC + GM, 2250 + 36,000 kg/ha), medium RSC + GM (MRSC + GM, 4500 + 36,000 kg/ha) and high RSC + GM (HRSC + GM, 6750 + 36,000 kg/ha). Among the tested fertilizers, urea ($CH_4N_2O$) was the chemical fertilizer, and rapeseed cake was the by-product of Changsha local rapeseed after being pressed, N: 6.24%; $P_2O_5$: 2.65%; $K_2O$: 1.71%. Green manure is a green manure variety chafei 1$^\#$ bred by Hunan Tea Research Institute.

Each treatment has three replicates with randomized block design. In the present study, the soil samples were treated with various combinations of organic and inorganic fertilizers. After the application the soil samples were collected and the nutrient content and microbial population in the samples were analyzed.

### 2.2. Soil Physicochemical Analysis

In May 2018, soil samples from 0–20 cm soil layers of tea garden were collected by 5-point sampling method in each experimental plot. The soil samples for all the treatments were collected and stored at −80 °C for further analysis. Soil pH and available nitrogen (AN) were measured by a combination pH electrode (soil: Water, 1:2.5 *w/v*) and alkaline hydrolysable method [11], respectively. Total nitrogen (TN) and total organic carbon (TOC) were measured by Kjeldahl digestion [12], total potassium (TK) and total phosphorus (TP) were determined by NaOH melt flamer and sodium carbonate method, respectively [13,14]. Available phosphorus (AP) and available potassium (AK) were extracted by hydrochloric acid and flame photometry, respectively [15].

### 2.3. DNA Extraction, Amplification and Pyrosequencing Analysis

The microbial consortia of organic and inorganic fertilizer treated soil samples were analyzed by Hiseq sequencing approach. Bacterial and fungal DNA was extracted from the soil samples by using OMEGA E.Z.N.A soil DNA kit (Norcross, GA, USA) according to the manufacturer's instructions [16] and the DNA quality was detected by 1.2% agarose

gel electrophoresis. To identify the fungal diversity in the tea garden soil, the extracted sequences were amplified with primer pairs (5′-CTTGGTCATTTAGAGGAAGTAA-3′) and (5′-GCTGCGTTCTTCATCGATGC-3′) that's amplify ITS1 and ITS4 regions of internal transcribed spacer (ITS). Meanwhile, primers pair, 515F (5′-GTGCCAGCMGCCGCGGTAA-3′) and 806R (5′-GGACTACHVGGGTWTCTAAT-3′) were amplify V3-V4 region of the 16S rRNA to detect the bacterial diversity in the tea garden soil. The PCR was conducted with the following cycling parameters: 95 °C for 2 min as initial denaturation, 35 cycles of 95 °C for 30 s, 55 °C for 1 min, 72 °C for 1 min and a final extension at 72 °C for 10 min. The purified PCR products were used for DNA library construction using TruSeq® DNA PCR-free sample preparation kit. After assessing the DNA quality, the DNA library was sequenced on Illumina HiSeq platform provided by Beijing Genomics Institute (Shenzhen, China).

### 2.4. Bioinformatics and Statistical Analysis

All 250 bp pair-end reads were connected using Connecting Overlapped Pair-end software (COPE, V 1.2.1) [17] after removing barcodes and adaptor sequences. The quality of the raw tags were controlled by trimming low quality sequences using Qiime V1.9.1 (http://qiime.org/scripts/split_libraries_fastq.html (accessed on 14 May 2018)) [18]. Finally, the effective tags were obtained after detecting and removing chimera by UCHIME Algorithm (http://www.drive5.com/usearch/manual/uchime-algo.html (accessed on 1 October 2017)) and Unite database (https://unite.ut.ee/ (accessed on 5 April 2018)) respectively [19,20]. The effective tags were used for further analysis. Effective tags were clustered into operational taxonomic units (OTUs) by using Uparse software (http://drive5.com/uparse/ (accessed on 1 October 2017)) [21]. The typical OTUs were annotated by blast analysis using Qiime software and Unit database (https://unite.ut.ee/ (accessed on 5 April 2018)) [22]. Multiple sequence alignment was conducted by MUSCLE V3.8.31 (http://www.drive5.com/muscle/ (accessed on 5 April 2018)) [23]. Community richness and diversity indices such as Chao 1, Shannon diversity and Simpson diversity indices were calculated by Qiime software. Diversity differences between samples were assessed by T test and Wilcox test. The Spearman's rank-bacteria correlation was used to investigate the correlation between the soil properties and the abundant genera. Differences of multiple comparisons were detected by R software with Tukey and Wilcox of agricolae package (https://www.r-project.org/ (accessed on 5 April 2018)) for abundance and diversity of bacterial and fungal species. Soil physicochemical properties analysis was conducted using SPSS 23.0 software package and the significant differences were assessed by One-way ANOVA analysis. $p < 0.05$ was considered as significant difference.

## 3. Results
### 3.1. Soil Physicochemical Analysis

In zero fertilizer application tea garden, the soil pH was significantly higher than in the fertilized tea garden soil. Particularly, the alkali-hydrolyzable nitrogen (AHN) was significantly higher in the fertilizer applied tea garden soil compared to zero fertilizer applied tea garden soil. Besides, there is no significant differences observed in AK, TN, TK and organic matter (OM) contents in CK and fertilizer applied tea garden soil samples though TP and AP contents were higher in MRSC + GM treated tea garden soil (Figure 1).

### 3.2. Abundance and Diversity of Bacterial and Fungal Species

According to Hiseq sequencing data on the phylum level, *Acidobacteria*, *Proteobacteria* and *Actinobacteria* are the most abundant bacteria, and *Ascomycota*, *Zygomycota* and *Basidiomycota* are the most abundant fungi (Figure 2). The relative abundance of bacterial species was higher in organic fertilizer applied tea garden soil than inorganic or zero fertilizer treated tea garden soil samples except *Acidobacteria* which was lower in RSC and MRSC + GM treatment. Specifically, *Chloroflexi* bacterial species population was higher almost 50% in organic fertilizer (RSC) treated soil samples followed by other bacterial species. However, the abundance of *Firmicutes* was lower in RSC applied soil than

other fertilizer treatments. For the fungi, very little *Basidiomycota* population was observed in MRSC + GM treated tea garden soil.

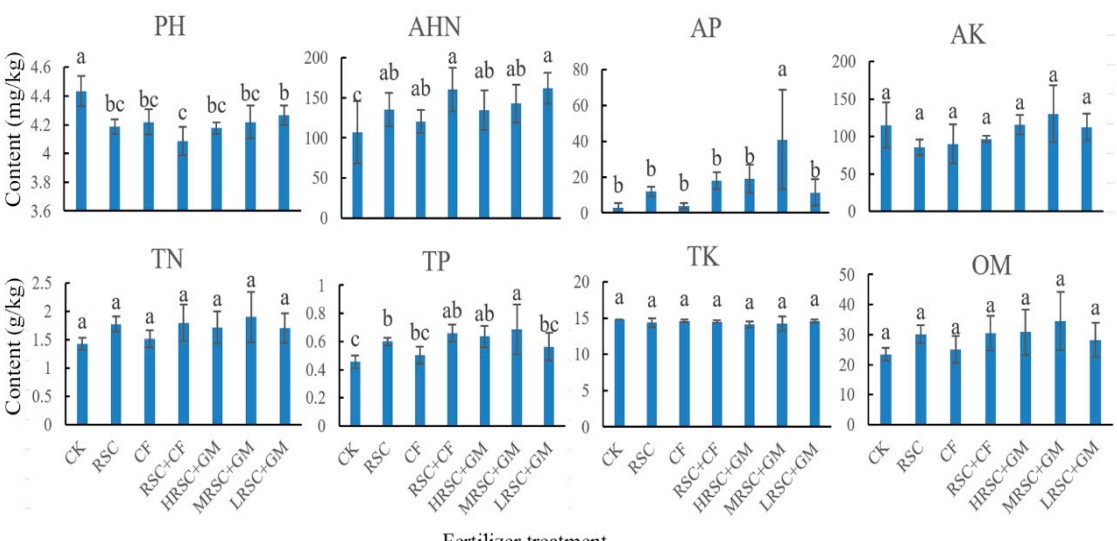

**Figure 1.** Soil physico-chemical properties from tea orchards with different fertilizer application.

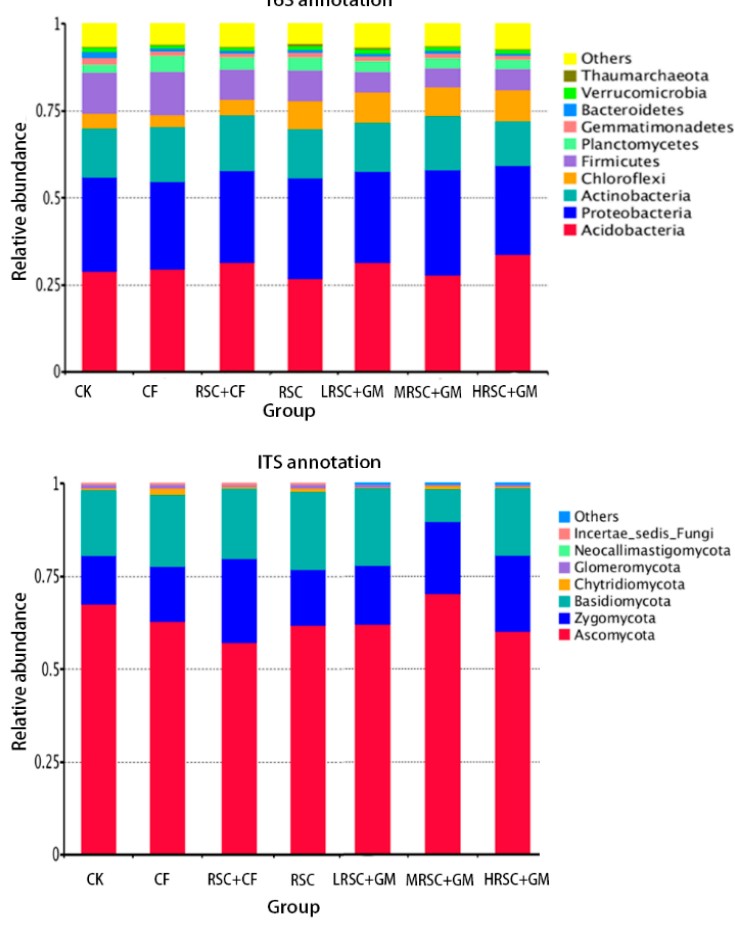

**Figure 2.** The abundant bacteria (16S rRNA annotation) and fungi (internal transcribed spacer (ITS) annotation) in soil of different fertilizer applied tea garden soil.

### 3.3. Abundance and Diversity of Microbes on Genus Level

As shown in Figure 3, the significant level microbial population differences were observed between the fertilizer applied and zero fertilizer applied tea garden soil. In control tea garden soil, *Haliangium*, *Sphingomonas*, *Gemmatimonas* and *Candidatus Solibacter* are the dominant microbes. In inorganic fertilizer applied tea garden soil (CF), *Acidothermus* and *Actinospica* are the dominant microbial species. In organic fertilized soil (RSC), *Mizugakiibacter*, *Kitasatospora* and *Mycobacterium* were the most abundant microbes. In contrast, the abundance of microbes significantly changed in integrated application of RSC with different manures.

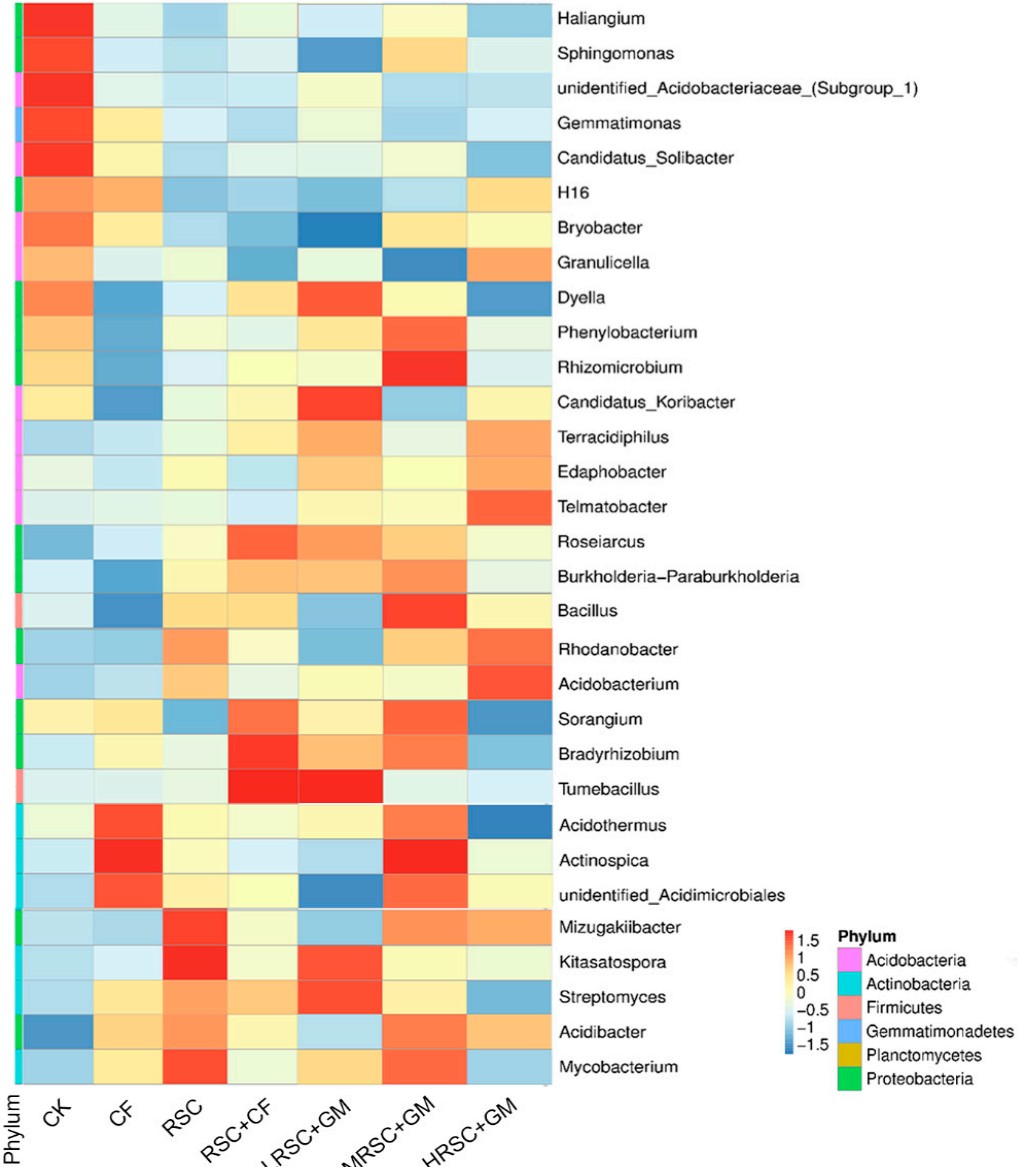

**Figure 3.** Top abundant microbes in different fertilizer applied tea garden soil on genus level.

### 3.4. Correlation between Bacterial and Fungal Communities with Soil Properties

The results revealed that the *Acidibacter* was positively correlated with AP and TP. *Acidothermus* was positively correlated to soil pH, but had negative correlation with AP, TN, TP and TOC (Figure 4). The Figure 5 showed that the *Trichoderma* spp., was positively correlated to AK. *Myrmecridium* had positive correlation with soil pH and AK, but negative correlation to TP and the ratio of total organic carbon to total nitrogen (CN). However,

*Climacodon* had negative correlation with pH and AK, but positive correlation with TP and TOC. In addition, *Khuskia* showed positive correlation with pH, but negative correlation with TN and TP. Therefore, these fungal communities may be sensitive to the change of soil properties of tea garden.

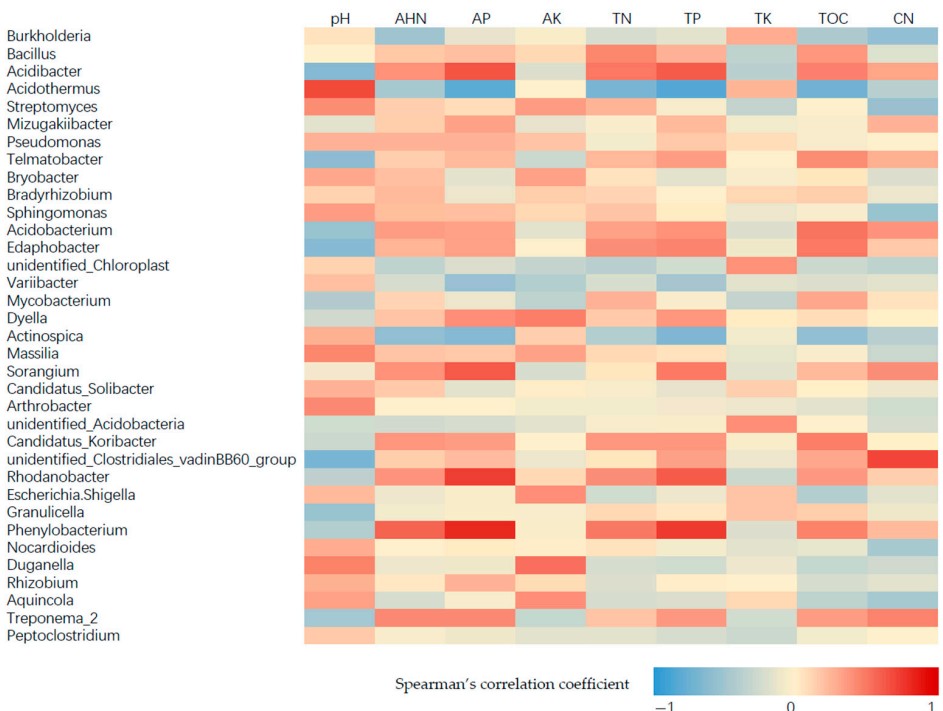

**Figure 4.** Spearman's correlation between bacterial communities and soil properties by different fertilizer treatment.

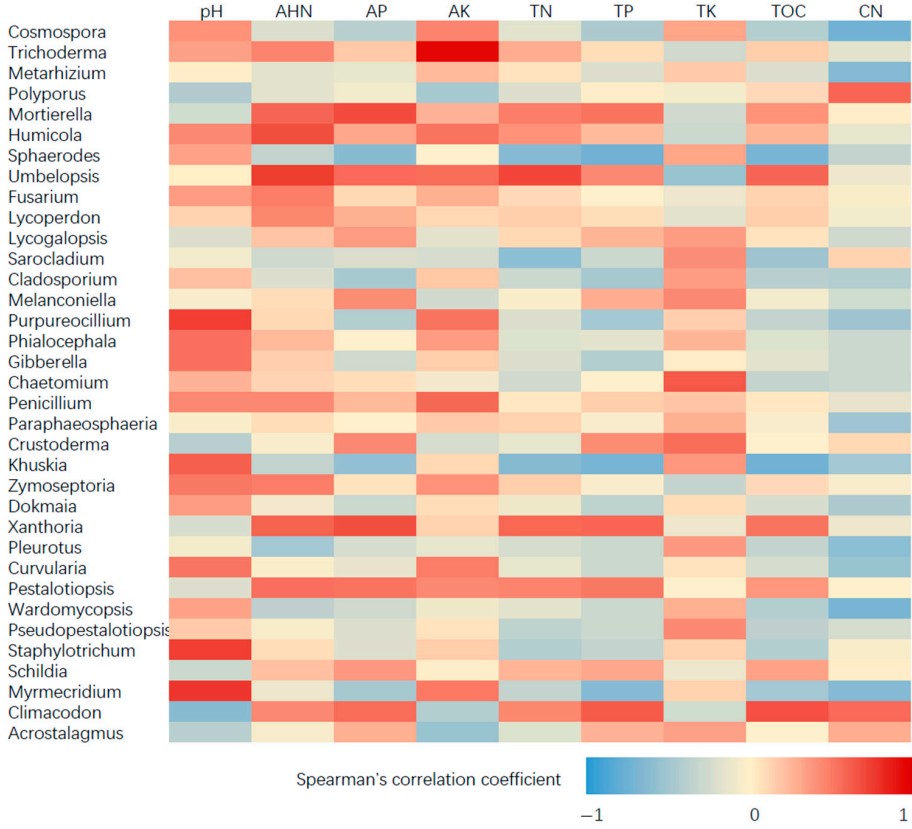

**Figure 5.** Spearman's correlation between fungal communities and soil properties by different fertilizer treatment.

## 4. Discussion

The obtained data suggested that the strategic management of soil organic inputs could greatly affect the soil fertility of the tea garden. The influence of microbial consortia on the productivity of the plantation crops along with nutrient content of the soil is an interesting area of plantation crop research in recent days. Several agricultural crops and their associated microbial populations studied earlier indicate that some special group of microorganisms are associated with a particular group of plants that directly influence the growth of the crop plants and the tolerance to biotic and abiotic stresses on various aspects [24].

A few studies have been reported [6,25] on the effect of fertilizer application on tea plantation correlated with microbial communities and soil biochemical properties. In the present study the influence of different organic and inorganic manures on the growth of soil microbial consortia was investigated. The obtained results suggest that *Acidibacter* and *Acidothermus* are important indicators for soil nutrient properties. This study showed the effects of different fertilizer application on soil properties and microbiota. The results indicated that fertilization increase the content of AHN but decrease pH, organic fertilizer slightly increase the content of OM, MRSC + GM had a significant impact on TP and AP of soil. Besides, significant level of microbial abundance and diversity were observed in different fertilizer application. Correlation analysis revealed that bacteria such as *Acidibacter* and *Acidothermus* and the fungi including *Myrmecridium*, *Climacodon* and *Khuskia* were significantly associated the soil properties (pH, TOC, TN, C/N, TP, AP and AK), which suggests that these microbes could be the indicators for soil nutrition status.

The increase in duration of continuous cropping years gradually altered soil chemical properties. Total N and C, exchangeable $NH^{4+}$-N were decreased. However, fertilizer practice may improve the quality of the tea orchard [26]. This study showed that organic fertilization could slightly increase the content of soil organic matter, AHN, TP and AP compare to no fertilizer and chemical fertilizer treatments. The chemical properties such as AP and TP were the most abundant elements compared to other fertilizer practices, which indicated that MRSC + GM could improve the TP and AP content in soil of tea garden. In inorganic fertilizer applied tea garden soil (CF), *Acidothermus* and *Actinospica* are the dominant microbial species. In organic fertilized soil (RSC), *Mizugakiibacter*, *Kitasatospora* and *Mycobacterium* were the most abundant microbes. In contrast, the abundance of microbes significantly changed in integrated application of RSC with different manures. Among all fertilizer treatments, MRSC + GM might be a potential fertilizer to improve soil microbial abundance, diversity, carbon and nutrient availability of tea garden.

Different fertilizer application has changed the abundance and structure of the microbiota in different soil ecosystem by different fertilizer practices. Of the physicochemical property measures, the Redundancy analysis showed that TOC had a significant effect ($p < 0.05$) on bacterial community in tea orchard. *Proteobacteria*, *Acidobacteria*, *Chloroflexi* and *Actinobacteria* were the dominant phyla in fertilizer soils, which was in accordance with the results of [8,27]. The results showed that *Proteobacteria*, *Acidobacteria*, *Chloroflexi* and *Actinobacteria* in the soil bacterial flora had strong growth ability in tea garden soil. Compared to other fertilization treatments, MRSC + GM increased the relative abundance of *Acidobacteria*, *Proteobacteria* and *Actinobacteria* in the soil. *Acidobacteria* were the most abundant phyla in soil but with very low availability [28]. *Proteobacteria* and *Acidobacteria* can promote the degradation and decomposition of straw in soil [29–31]. At genus level, *Acidibacter* and *Acidothermus* were associated to soil nutrient properties, and both bacteria were enriched in MRSC + GM treated soil. Thus, both bacteria were important to maintain the soil environment of tea plants. There are many bacteria which affect the nutrient in soil [32,33]. For examples, the extensively examined *Rhizobium* is a symbiotic bacterium of leguminous plants. $N_2$-fixation is the vital role of Rhizobia for plants. The $N_2$ fixed by Rhizobia in legumes is also beneficial to the intercrops via direct transfer of biologically fixed N. Thus, legumes were usually used as green manure planted in tea garden. Previous studies have identified that *Acidobacterium* occurred in diverse environments as a dominant

bacterial group [34,35]. The use of the measured physicochemical properties for fertility index calculation allows us to decipher the effects of soil fertility on the bacterial and fungal community in tea plantation soils.

## 5. Conclusions

Based on 3-year uninterrupted field experiment, dynamic effect was understood well about organic amendments supplementation on soil quality, which provides scientific basis for agriculture practice. Our field experiments showed that fertilizer treatments had no significant effect on TK. Compared with complete chemical fertilizer and no fertilizer application, organic fertilizer slightly increase the content of soil OM, TN and TP, significantly increased AHN and AP, but slightly reduced soil pH. In addition, our results found that bacteria such as *Acidibacter* and *Acidothermus* and the fungi including *Myrmecridium*, *Climacodon* and *Khuskia* were significantly associated the soil properties (pH, TOC, TN, C/N, TP, AP and AK), which suggests that these microbes could be the indicators for soil nutrition status. Among all fertilizer treatments, MRSC + GM might be a potential fertilizer to improve soil microbial abundance, diversity, carbon and nutrient availability of tea garden.

## 6. Patents

This section is not mandatory but may be added if there are patents resulting from the work reported in this manuscript.

**Author Contributions:** Conceptualization, methodology, investigation, software, validation, resources, data curation, writing—original draft preparation, H.F., H.L.; resources, formal analysis, data curation, writing—original draft preparation, P.Y., H.M. and J.L.; investigation, P.Z., Y.W.; methodology, data curation, writing—review, Q.M., A.J., K.T. and X.C.; conceptualization, supervision, writing—review, editing and funding, X.L.; supervision, funding acquisition, G.G. All authors have read and agreed to the published version of the manuscript.

**Funding:** This research was funded by National Key R&D program (2016YFD0200900), the Natural Science Foundation of Jiangsu Province (BK20200554), the National Natural Science Foundation of China (31800590), Special Fund for Agro-scientific Research in the Public Interest (201503110), the Nanjing special plan for Shangluo leading industry demand (20201105), the Open Fund of Henan Key Laboratory of Tea Plant Comprehensive Utilization in South Henan (HNKLTOF2020002), the China Earmarked Fund for Modern Agro-industry Technology Research System (CARS-19) and Expert Workstation of Yunnan Cuigong Tea Co., Ltd. (Zhaotong, China) in Zhaotong (2020ZTYX07).

**Institutional Review Board Statement:** The study was conducted according to the guidelines of the Declaration of Helsinki, and approved by the Institutional Review Board of Sustainability Editorial Office.

**Informed Consent Statement:** Informed consent was obtained from all subjects involved in the study.

**Acknowledgments:** We thank Yuehua Ma (Central laboratory of College of Horticulture, Nanjing Agricultural University) for using PCR (T100, Bio-rad, Hercules, CA, USA).

**Conflicts of Interest:** The authors declare no conflict of interest.

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
