# Peer review of "Integrated Application of Rapeseed Cake and Green Manure Enhances Soil Nutrients and Microbial Communities in Tea Garden Soil"

_sustainability, doi:10.3390/su13052967_

Round 1

Reviewer 1 Report

The manuscript submitted by Fu et al. investigates the effect of different organic fertilizers based on rapeseed cake and green manure on the nutrient parameters and microbiota of tea garden soil. The subject is important from a sustainable agriculture point of view, but the paper has some serious flows that should be corrected. 

First of all, the methodology is not very clearly described. The authors should give more details about the experimental design, such as the surface for each parcel, number of planted plants and especially the period after which the soil samples were collected and in which season. 

Second, the authors mention (lines 58-60) that they also analysed the biochemical components, productivity and quality of tea, but this information is completely missing from the manuscript. These data would have been indeed useful to sustain the conclusions from the discussion section. The authors should clarify this aspect and write the paper and conclusions in accordance with the data they have.

The observation that the soil nutrients increase and that there are some positive changes in beneficial microbiota does not guarantee better tea yield and quality. In the absence of relevant data, the authors should at least relate to known information from other studies, especially with respect to microbiota influence on tea production and quality.

The discussion part should be extended to include also the observations (positive or negative) from the other treatments, not just MRSG+GM. Moreover, the correlations between the effects of different nutrients, microbiota and the changes induced by the microbiota in soil and plants should be presented in a more straightforward way. The authors should also be careful with verb tenses in the discussion, because it is not clear what information is from literature and what is derived from the results. Use present tense to what is already known from other reports and indicate clearly the conclusion derived from this study. Indicating the Figures where the information appears would be helpful for the readers to follow the text.

Please check the English style, there are some flaws that need to be addressed.

Please see also the PDF attached, for these observations and some minor issues. 

Reviewer 2 Report

Referee Report for the Manuscript ‘Integrated application of rapeseed cake and green manure enhance soil nutrients and microbial communities in tea garden soil' submitted to the Sustainability by Haiping Fu, Huan Li, Peng Yin, Huiling Mei, Jianjie Li, Pinqian Zhou, Yuanjiang Wang, Qingping Ma, Anburaj Jeyaraj, Kuberan Thangaraj, Xuan Chen, Xinghui Li, Guiyi Guo

The manuscript analyses the potential effects of fertilizers on the abundance and the structure of microbiota in tea garden soil, soil properties and tea productivity and quality.
Potentially, the manuscript provides an important contribution to the research area.

The Conclusion Section could be conceptually developed to complete fully the manuscript.

Reviewer 3 Report

This manuscript reports on an effort to investigate the effect of organic amendments (RSC+GM) and inorganic fertilizers on soil nutrient dynamics and microbial abundance in tea garden soil. Author has analyzed the soil chemical properties as a function of various organic and inorganic fertilizing treatments and further, explored the relationship between these soil chemical properties vs soil microbial abundance. Manuscript needs revision at current stage. Points to address during revision follow:  

General comments.

Experimental design and fertilizing treatment (section 1; Line 65) should be presented in detail such as cultivation practices, amount/type/method of inorganic fertilizer amendment, type of plants used for green manure, characterization of organic inputs (SRC and GM), basis for choosing three RKC rate with one consistent GM rate etc. These details are crucial to explain the potential of various treatments and their complex interactions on soil nutrient status and microbial abundance. Statistical analysis needs to be specific with appropriate model for each of the analysis being performed (see specific comments below). Discussion section needs to be revised and fine tuned to provide the specific mechanistic explanation of the obtained results. Author suggested MSRC+GM could be the preferred fertilizer for tea garden based on AP and TP availability which has also shown positive relationship with microbial activities. However, study lacks the efficiency of MSRC+GM on available N and other secondary and micronutrients, which is critical while recommending the fertilizers. Thus, author need to clarify whether N and other fertilizers could be supplied by MSRC+GM alone or needs to be supplied through external source based on the current findings (soil and microbial status) and discussing the findings with other relevant literatures. Further, the applied inorganic fertilizer is not known in this study making it difficult to identify the amount of nutrients that needs to be supplied and bioavailable through organic amendments in tea garden soil. Author may also add ‘conclusion section’ summarizing the main findings, their implications and research gaps as a way forward!

Last but not the least, there are grammatical error, which needs to be considered and corrected throughout the manuscript to ensure readability.

Specific comments;

Line 26. Remove ‘physical’ as you have not assessed any soil physical properties in the study. Further, list out which chemical properties were significantly correlated along with their respective correlation coefficient.

Line 29. Specify the soil nutrients? Soil nutrients also include nitrogen and other secondary and micronutrients, which is not investigated or discussed in this study.

Line 48-49. Mention the N inputs rate in range practiced in the tea garden.

Line 58. For how many years.?

Line 67-68; Discuss briefly about the cultivation practices that was performed in the experimental field from where soil sample was collected.

Line 69. Mention the rate of chemical fertilizer and rape seed cake (RSC treatment). Which plant was used for green manuring? Also discuss briefly about the application method of these fertilizers in each of the experimental plot. Did you characterize the organic fertilizers (rapeseed cake and green manure), if not, then you may need to clarify, how you select the rate for RSC and green manure and classify them as low, medium and high in tea garden soil.  

Line 73-74. You may move this sentence to next section (2.2. soil physiochemical analysis) and discuss briefly about the soil sampling procedure as well (such as soil collection methods and soil depth).  

Line 79: “and” is redundant

Line 114: Did you mean “tukey” test, or I misunderstood?

Line 116. Here you have mentioned SPSS for all statistical analysis and in previous line (114) you have used R package for t-test and multiple comparison with tukey test. You need to specify the software (SPSS or R) for each analysis with their respective model that you have performed.

Line 120-122: This sentence is better fit in material and method section.

Line 122-124: Too early to state this. Maybe you can discuss this later in discussion section after you present your result/data first, in this section.

Line 132. In figure 1 caption, you may remove the abbreviation of different soil chemical/nutrient parameters as you have already stated them in methods and result section. Same comments for figure 4 and 5. 

Line 126: “in” redundant  

Line 136. You have already mentioned this in line 56-58. Further, this sentence is more appropriate in materials and method section.

Line 140-141: Replace “had” with “was”. You may present the magnitude of abundance in organic treatments (for e.g. by how much % higher compared with CK and CF plot). Further, figure 2 does not show higher abundance of all bacterial species in organic fertilizer treatment. For example, bacterial abundance (acidobacteria) in RSC and MSRC+GM seems similar or even lower compared with CK and CF treatment. I would suggest being specific highlighting only those bacterial spp which has shown significant variation among the treatments.

Line 143. This is also true for acidobacteria?

Line 161-162. I would suggest keeping all the analysis that you have performed under “bioinformatics and statistical analysis section” in line 99-117.

Line 163. Positively instead of ‘positive’.

Line 164-165. You may reserve this for discussion section.

Line 182. I did not see OM (organic matter) in the figure.

Line 167. What is difference between Ak and sAK? Further, you did not mention about sAK in method/result section of soil physiochemical analysis. If there is a difference, then, you need to mention sAK analysis in method and present its properties in result section.

Line 187-189. Provide reference.

Line 193. It is already mentioned in Introduction section (line 56-60). Here, you need to discuss your findings.

Line 201. Point out for which soil chemical properties (this is not true for all soil properties). Further, discuss the potential role of these microbes in making nutrient available (positively correlated) and unavailable (negatively correlated) in the tea garden.

Line 203. “the” redundant. Further, the sentence is not clear “agriculture production gradually decreased the content the soil chemical properties”. You may rewrite making it clear.

Line 208. You have applied GM in consistent amount in all GM amended treatments. Thus, there could be the main effect of RSC rate or interaction effect between different RSC rate and GM on available AP and TP. Please discuss your finding with other relevant mechanistic studies to support that the medium rate RSC + GM have beneficial effect on AP and TP, and not with low or even higher RSC rate!

Line 212: Spell out RDA. As mentioned above, for multivariate statistics, mention the approach with their respective model that have been performed in this study in the material and method section (bioinformatics and statistical analysis part).

Line 224 – 228. But in your study, relationship between rhizobium and available N was not investigated. Further, no relationship between rhizobium and total N content was observed (figure 4).

Line 234-235. From your study, you may specify optimal P fertilizers application in tea garden soil.  In my opinion, you may also need to discuss  other essential nutrients such as N and other secondary/micro nutrients while recommending optimal fertilizers; how much of these nutrients could be supplied from MSRC+GM or whether you may need to supply from external source. I would suggest discussing them based on your soil chemical properties and microbial abundance data and their complex interactions to make N and other nutrients bioavailable taking a reference from other relevant studies. In other hand, If nutrients (N and others) are not crucial to apply in teagarden, then, you may also need to mention this at some point in discussion section, highlighting the importance of P fertilizers in tea garden soil.

Round 2

Reviewer 1 Report

The authors managed to improve the manuscript by providing additional information, removing the information not sustained by the results and presenting and discussing the results more clearly. The language, grammar and style still need some improvements. I have one more minor comment related to the treatment period. At line 61 the authors mention 5 years, while at line 299, the authors claim that the field experiment took 3 years. It is confusing, please clarify. 

Reviewer 3 Report

Author has revised the manuscript responding the comments and feedback specifically. Still, language and grammatical errors needs to be reconsidered and corrected throughout the MS. I have minor comments as follows: 

  • In line 74, please list the source of chemical fertilizer that was used. However, in line 77, you have mentioned that urea was the chemical fertilizer. What about P and K? If CF treatment include only N (urea) then present this making it clear.
  • Line 86. mention the unit (0-20 cm?)
